# Recent Advances in Metal-Based Nanoparticle-Mediated Biological Effects in *Arabidopsis thaliana*: A Mini Review

**DOI:** 10.3390/ma15134539

**Published:** 2022-06-28

**Authors:** Min Geng, Linlin Li, Mingjun Ai, Jun Jin, Die Hu, Kai Song

**Affiliations:** 1College of Food and Biology, Changchun Polytechnic, Changchun 130033, China; baobeibeilj@163.com; 2School of Life Science, Changchun Normal University, Changchun 130032, China; lin1805626597@163.com (L.L.); aimingjun789@163.com (M.A.); junking1018@163.com (J.J.); hudie18784775519@163.com (D.H.); 3Institute of Science, Technology and Innovation, Changchun Normal University, Changchun 130032, China

**Keywords:** metal-based nanoparticles, *Arabidopsis thaliana*, phytotoxicity, biological effects, expose

## Abstract

The widespread application of metal-based nanoparticles (MNPs) has prompted great interest in nano-biosafety. Consequently, as more and more MNPs are released into the environment and eventually sink into the soil, plants, as an essential component of the ecosystem, are at greater risk of exposure and response to these MNPs. Therefore, to understand the potential impact of nanoparticles on the environment, their effects should be thoroughly investigated. *Arabidopsis* (*Arabidopsis thaliana* L.) is an ideal model plant for studying the impact of environmental stress on plants’ growth and development because the ways in which *Arabidopsis* adapt to these stresses resemble those of many plants, and therefore, conclusions obtained from these scientific studies have often been used as the universal reference for other plants. This study reviewed the main findings of present-day interactions between MNPs and *Arabidopsis thaliana* from plant internalization to phytotoxic effects to reveal the mechanisms by which nanomaterials affect plant growth and development. We also analyzed the remaining unsolved problems in this field and provide a perspective for future research directions.

## 1. Introduction

Metal nanoparticles (MNPs) are widely used in biosensors, medical imaging, diagnostic and therapeutic materials, antimicrobial agents and drugs, chemical catalysis, optoelectronics and other areas due to their unique physical and excellent chemical properties [1,2,3,4,5,6]. However, during their production and recycling process, MNPs are inevitably leaked into the environment and may become exogenous stimulation for plants.

MNPs have been extensively studied for their effects on plant growth and development. For instance, TiO_2_ NPs were found to promote seed germination in tomato, onion and radish, Cu NPs concentration-dependently inhibit seedling growth and root growth in both mung bean and wheat, and Al_2_O_3_ NPs were observed to have no pharmacological effect on the root elongation of *Arabidopsis*, radish, rape, ryegrass, lettuce or cucumber [7,8,9].

Soil has gradually become a significant reservoir for MNP deposits in the environment. Several previous studies have shown that the number of residual nanoparticles in the soil of certain regions could reach up to 1.9–865 mg/kg [10]. These MNPs were absorbed by plant roots and leaves and transported to other tissues via the plant’s vascular system [11,12,13]. However, most studies found that MNPs have a low rate of internalization [14,15,16]. For example, the translocation of Ce from the roots to the stems was shown to be around 1.44% and 1.79% [17]. Nevertheless, these internalized MNPs still pose a risk due to their ability to translocate, accumulate and even transform within the plants to interact with biomolecules, leading to changes in the morphological characteristics, physiological responses and growth of plants at a molecular level. Therefore, it is necessary to thoroughly evaluate the interactions between nanoparticles and plants.

Due to the limited number of plant species and technical means used in the existing literature, observations of the combination of different plants and MNPs have reached no unified conclusions. Previous works have shown that *Arabidopsis thaliana* is an ideal plant that can be used to model the impact of environmental factors on plants because the ways in which they develop, reproduce and respond to adapt to these factors are representative of those of other plants. For this purpose, this review covers the recent studies on several important types of MNP-mediated biological effects in *Arabidopsis* over the past decade. To the best of our knowledge, this is the first review of the biological effects of Arabidopsis based on metal nanomaterials; additionally, it covers the uptake and transport of MNPs in *Arabidopsis*, and the corresponding stress responses of *Arabidopsis* to these MNPs at the morphological, physiological and molecular levels to provide the theoretical basis for assessing the effects of nanoparticles on the environment.

## 2. Absorption and Transport of MNPs in *Arabidopsis thaliana*

Based on studies performed to assess the ecological safety of nanomaterials, it is generally believed that nanoparticles exposed to the surface of plants can be attached to tissue surfaces and hinder the transmission of water, nutrients and ion exchange. Some hydrophilic nanoparticles have been found to cross plants’ cell walls and accumulate between cell walls and cell membranes or between cell walls of adjacent cells, indicating a potential plasmatic exosomal transport mode for nanoparticles in plant tissues. Despite limited evidence showing that intact plant roots can absorb and translocate nanoparticles [18,19,20,21], there are still controversies surrounding this issue [22,23].

### 2.1. Absorption and Transport of Monometallic Nanoparticles in Arabidopsis thaliana

Geisler-Lee et al. tested different sizes of Ag NPs (20, 40, 80 nm) in a hydroponic growth media using different microscopy methods to study the effects of Ag NPs’ toxicity in *Arabidopsis* root tips. They found that Ag NPs were absorbed and gradually accumulated in the root tips, from the marginal cells to the root cap, epidermis and columella, and then penetrated the initial part of the root meristem (Figure 1a). At low concentrations, smaller Ag NPs accumulated more than larger ones, while at high concentrations, the opposite occurred [24,25]. Ag NPs were first absorbed by underground tissues (primary root and lateral roots) and then transferred to aboveground parts (stems, leaves, flowers, etc.) where they tended to influence the growth and development of *Arabidopsis thaliana*. They appeared to accumulate in the plastid exosomes of root tissues while only a tiny fraction was transported to aboveground tissues. In the places they accumulated, i.e., on the surface of bare plant roots and leaves, they demonstrated a low internalization rate. It was found that the particle size of Ag NPs in plant tissues was larger than their initial diameter, suggesting that the internalized Ag NPs no longer existed as intact individual particles but rather appeared to aggregate and biotransform in the plants [26]. In contrast to observations made for Ag NPs, Au NPs (60 nm) tended to remain unaggregated after being absorbed by *Arabidopsis* roots. Yeonjong Koo et al. compared leaf acoustic signal distributions from Arabidopsis leaves exposed to media with high (2.4 × 10^10^ NP mL^−2^) or low (4.8 × 10^8^ NP mL^−2^) GNP concentrations. The high GNP concentration increased the percentage of the leaf surface area, but regardless of concentration, nearly all the signals remained in the 90–200 mV amplitude range. A lack of high-amplitude signals suggests that GNPs did not aggregate in plants (Figure 1b) [27]. Thus, it seems that in addition to the changes in morphology and concentrations of monometallic nanomaterials that occur in *Arabidopsis*, other factors also affect the state of MNPs in plants.

The surface charge of nanoparticles is generally assumed to be a key factor affecting their uptake and translocation. Using DF-HSI and nano-CT, Astrid et al. observed that negatively charged nanoparticles were transported along plastid exosome in *Arabidopsis* while positively charged nanoparticles uptake occurred to a small extent, possibly through other processes, such as clathrin-mediated endocytosis, in the phytoplankton (Figure 2) [29]. However, Milewska-Hendel et al. modified the surface of AuNPs using polyethylene glycol (PEG) and branched polyethyleneimine (BPEI) and citrate to achieve neutral, positive and negative charges, as demonstrated by HRTEM analysis, which demonstrated that, regardless of the surface charge of Au NPs, they did not traverse the cell wall barrier of *Arabidopsis* root bark cells or root cap cells but were internalized by the protoplasm [30]. Although there seems to be some strong co-localization of Au NPs in root tips, it has not yet been possible to determine whether Au NPs are adsorbed on or accumulated in the roots.

### 2.2. Absorption and Transport of Metal Oxide Nanoparticles in Arabidopsis thaliana

The use of zinc oxide nanoparticles (ZnO NPs) as Zn fertilizer has been shown to be effective for correcting Zn deficiency in soils [31]. However, it has also been shown that ZnO NPs may dissolve rapidly once they are released into the soil, releasing Zn ions, and may lead to a far higher concentration of Zn than expected [32]. In plants, Zn homeostasis is mediated through transporter proteins involved in the intracellular acquisition of Zn, mobilization and sequestration [33]. The *Arabidopsis* transporter proteins AtZIP4, AtZIP9 and AtZIP12 are involved in the acquisition of Zn from roots and subsequent mobilization to aerial tissues, while AtHMA3 and AtHMA4 mediate root-to-crown Zn transport [34,35]. Prakash et al. observed *Arabidopsis* seedlings after treatment with ZnO NPs under fluorescent labeling. They detected an intense green fluorescence in the primordial root tip region, primordial lateral root junctions and aboveground root junctions, but ZnO NPs treatment resulted in Zn accumulation only in the root apex and root–shoot junctions, whereas Zn ion treatment caused a root-to-shoot uptake and translocation of the element (Figure 3) [36].

In experiments where *Arabidopsis* was exposed to 5–40 mg/L of CuO NPs, the Cu content in *Arabidopsis* roots was significantly increased compared to the Cu content in *Arabidopsis* stems and leaves. Additionally, while the transfer rate of CuO NPs from root to shoot was found to be low (1.1–2.8%), under the same conditions, that of Cu^2+^ occurred at a higher rate (10.8%), indicating a weak transport capacity of CuO NPs (Figure 4) [37]. Wang et al. exposed *Arabidopsis* to 50 mg/L of CuO NPs and found that the Cu contents in the roots were significantly higher than those in leaves, flowers and harvested seeds in the investigated ecotypes of *Arabidopsis*. In all the tissues tested, the Cu contents were significantly higher after exposure to 50 mg/L of CuO NPs than exposure to 0.15 mg/L of Cu^2+^, indicating that a large number of CuO NPs were transformed and transported as Cu^2+^ in *Arabidopsis* [38]. Thus, based on metal oxide nanoparticles’ solubility, comparing the effect of the nanoparticles themselves with that of a single metal ion is important to determine the extent of their internalization in plants.

Unlike highly soluble MNPs, TiO_2_ NPs are difficult for plant roots to absorb due to their low solubility. In addition, titanium also plays a key role in plants as it stimulates the production of more carbohydrates and helps in encouraging growth and the rate of photosynthesis. Ti/TiO_2_, widely used in the agricultural sector, exhibited both phytotoxic and positive effects on the size, concentration and plant species tested [39].

Although Ti elements are non-essential elements for *Arabidopsis thaliana* because their cell membranes lack corresponding transport receptors, Kurepa et al. found that TiO_2_ NPs (<5 nm) could be absorbed, translocated and distributed among the tissues and cells of *Arabidopsis* seedlings [12]. Via morphological and histological assessment of ultrasmall TiO_2_ NPs, García-Sánchez et al. observed that TiO_2_ NPs could enter *Arabidopsis* cells, accumulate in subcellular (including vesicular) locations such as the cytosol and root cell nuclei and further disrupt *Arabidopsis* microtubule dynamics [12,40,41]. This suggests that there are still other unknown ways and pathways for MNPs to enter *Arabidopsis*, and it would be helpful to further assess TiO_2_ NPs using traceable signals.

CeO_2_ NPs are a class of MNPs that tend to aggregate and precipitate in aqueous solutions due to their size and surface properties. In a study by Yang et al., the investigators introduced an agar curing medium to prevent the aggregation of CeO_2_ NPs, allowing them to be uniformly dispersed. It was found that the transport of Ce compounds by *Arabidopsis* grown in the agar medium behaved similarly to internalized CuO NPs in plants [42]. Ma et al. digested and analyzed *Arabidopsis* exposed at 0–1000 ppm CeO_2_ NPs by ICP-MS and observed measurable amounts of the elements in the root and stem tissues of *Arabidopsis*. However, the underlying mechanism of this transport is yet to be uncovered. Despite these observations, the accumulation and translocation of CeO_2_ NPs in plants seem to vary depending on the plant species. Birbaum et al. found that CeO_2_ NPs did not undergo translocation in maize, while Ce elements were found to be accumulated in plants such as alfalfa, cucumber and tomato [43,44,45,46].

### 2.3. Absorption and Transport of Other Metal-Based Nanoparticles in Arabidopsis thaliana

Given their promising water solubility and small size, quantum dots (QDs) were believed to be easily absorbed by plants; this was also confirmed in the recent study of pumpkin’s physiological responses to zinc oxide quantum dots and nanoparticles [47]. The experimental results for water-dispersible CdSe/ZnSe QDs showed no significant results [48]. Using confocal fluorescence microscopy, Navarro et al. found that water-soluble CdSe/ZnS QDs with carboxyl groups were strongly adsorbed to polar/charged root surfaces but could not enter the roots. Moreover, despite a 7-day exposure period, the plant cells remained impermeable to QDs, and therefore, QDs could neither be endocytosed nor passively or actively transported through the plant root system (Figure 5), suggesting the significant effect of the surface charge of nanoparticles on their uptake by *Arabidopsis*. In addition to the barrier created by the plant’s cell wall, when QDs are electrostatically adsorbed on the root surface, they form bulky agglomerates, which further impedes their entry as endocytosis cannot occur [49].

Taken together, the current body of literature suggests that although the uptake of most MNPs is associated with ion transporters on *Arabidopsis* root cell membranes, they have a low rate of internalization [50]. Small numbers of MNPs that are ingested or able to enter root cells via other routes are biotransformed into an ionic state and transported to other parts of *Arabidopsis*. Besides this, the importance of nanomaterials’ entry through stomata has also been extensively studied [51,52]. Moreover, the size, charge and growth media of nanoparticles affect the extent to which they are absorbed and transported.

## 3. Phytotoxic Effects of MNPs on *Arabidopsis thaliana*

The large surface area and small size of NPs are some of their desirable attributes that allow them to substantially ameliorate plants’ physiological processes. Nevertheless, the results derived from such research have not always been positive as NPs have been shown, in some cases, to negatively affect plants due to their potentially toxic nature [53]. Despite the low internalization and transfer rate of MNPs in *Arabidopsis*, most studies confirmed that MNPs have predominantly harmful effects on the plant. Table 1 summarizes the phytotoxic effects of different MNPs on *Arabidopsis*.

### 3.1. Toxic Effects at the Morphological Level

Growth potential, seed germination, biomass and leaf surface area are the commonly used parameters to assess morphological changes in plants and phytotoxicity [74,75,76,77,78]. Kaveh et al. found that exposure to Ag NPs (from 5 to 20 mg/L) resulted in reduced biomass of *Arabidopsis*, while Ag NPs at concentrations of 50 mg/L and 60 mg/L had a positive effect on root growth. In addition, the shapes of Ag NPs were also found to be an important factor influencing root growth [79]. Ag NPs with triangular and decahedral shapes promoted root growth, while spherical Ag NPs had no effects on *Arabidopsis* seedlings [80].

The number and length of *Arabidopsis* lateral roots have been found to be significantly reduced after treatment with different sizes and concentrations of Au NPs solutions, whereby smaller (10 nm) Au NPs negatively affected primary root growth but significantly promoted root hair growth (Figure 6) [61].

Although some studies have shown that ZnO NPs could significantly inhibit *Arabidopsis* growth and biomass accumulation, low concentrations of less than 20 mg/L are considered non-effective. Treatment at concentrations greater than 50 mg/L was found to significantly reduce the fresh weight and primary root length of *Arabidopsis* seedlings, decrease leaf size and yellowing and alter root structure [36,65].

Unlike other metal oxide nanoparticles, iron oxide nanoparticles (IONPs) are believed to interact with *Arabidopsis* in a charge-dependent manner as charged IONP could be detected in *Arabidopsis*’s root, leaf, flower and hornbeam tissues. Further, negatively charged IONP had a significant inhibitory effect on *Arabidopsis* seedlings and roots’ length, while positively and negatively charged IONP inhibited pollen viability, pollen tube growth and seed yield (Figure 7). These results indicate the detrimental effect of IONPs on the whole plant’s reproduction cycle. This phytotoxicity has also been found to be related to the concentration of IONPs, and the number of IONPs exposed to the plant [73].

The morphotropic effects of MNPs on *Arabidopsis* may also be related to the alteration of the porosity of the plant’s cell walls when stimulated under different concentrations of MNPs. While larger pores tend to promote the uptake of nutrients and MNPs from the soil, MNPs tend to simultaneously cause root clogging and impede the entry of nutrients.

### 3.2. Toxic Effects at the Physiological Level

Oxidative-stress-induced by CuO NPs or their released Cu^2+^ is the primary mechanism by which CuO NPs induce their phytotoxic effects [81]. In 2017, Ke et al. demonstrated that the phytotoxic effects of CuO NPs mainly originated from the nanoparticles themselves; dissolved Cu^2+^ contributed only a tiny fraction to the toxicity induced by CuO NPs. Upon 2 h of exposure to CuO NPs (10 mg/L and 20 mg/L), severe damage of *Arabidopsis* root cells was observed, whereas corresponding exposure with dissolved Cu^2+^ (0.80 mg/L and 1.35 mg/L) did not damage the roots and only exhibited partial phytotoxic effects after 12 days of exposure [67]. Moreover, studies have shown that *Arabidopsis* roots, leaves, flowers and harvested seeds treated with CuO NPs contained significantly higher Cu elements than CuO bulk particles (BPs) and Cu^2+^ treatment, demonstrating that CuO BPs cannot be readily accumulated and distributed in the plant (Figure 8), indicating that CuO NPs themselves have high mobility in *Arabidopsis*. In combined ethylene experiments, CuO NPs were observed to induce oxidative stress and inhibit growth by affecting the rosette size, biomass, chlorophyll content, lipid peroxidation, accumulation of oxygen species and cellular ultrastructure of *Arabidopsis* [38,68]. Interestingly, Jia et al. found that the Cys cycle affected the uptake and intracellular transport of CuO NPs. The activity and toxicity of CuO NPs were reduced by promoting the production of chelators while stabilizing the level of CuO NP-induced reactive oxygen species [69].

At low concentrations, CeO_2_ NPs can promote growth, but at high concentrations (1 g/L), they induce glutathione metabolism (oxidative stress response) and inhibit chlorophyll production and plant growth [42,82]. These changes in physiological levels are believed to be caused by the nanoparticles themselves, rather than by dissolved Ce^4+^, and are particle-specific [72].

Although it is generally accepted that oxidative stress (ROS) could be one of the direct sources of nanomaterials’ toxicity, relevant experiments showed that changes in ROS may not be the direct cause of the toxic effects of TiO_2_ NPs. TiO_2_ NPs are extremely easy to agglomerate in aqueous solutions due to their large specific surface area, high surface energy and severe lack of coordination, making it necessary to ensure the stability of TiO_2_ NP dispersions in all relevant plant studies. Interestingly, it was found that the presence of tetracycline (TC) could significantly reduce the accumulation of Ti^4+^ released by TiO_2_ NPs in branches and roots [71]. Further, *Arabidopsis* could also mitigate chloroplast oxidative damage caused by low doses of TiO_2_ NPs through autophagy [70]. These findings provide essential information needed to understand the interaction between metal-based nanoparticles and contaminants, such as antibiotics, in the plant systems.

Additionally, MNPs have been found to affect the photosynthetic efficiency of *Arabidopsis* in several other ways. For example, CuO NPs, CeO_2_ NPs and ZnO NPs can disrupt the energy transfer or oxidation from the photosystem to the Calvin cycle and reduce the gas exchange dynamics [83]. Further, ZnO NPs can also alter the photosynthetic core by releasing Zn^2+^ instead of Mg^2+^ in the chlorophyll center, and exhibit tissue specificity and concentration dependence. These observations corroborate experiments from phytohormone analysis [64,84]. Ag NPs, on the other hand, can accumulate in *Arabidopsis* leaves through the particles themselves to further disrupt the cystoid membrane structure, reduce chlorophyll content and inhibit plant growth [85].

Accumulating evidence from more detailed studies shows that the effects of MNPs on the photosynthetic system of *Arabidopsis* are complex. Sperdouli et al. exposed *Arabidopsis* leaves to CuZn NPs via foliar spraying and found that the photosystem II (PSII) function of young leaves was negatively affected, which could be attributed to the MNPs impeding the photosynthetic pathway by blocking the electron transport chain. In contrast, they observed a beneficial effect on PSII function in mature leaves and suggested that MNPs promoted the photosynthetic processes by improving light-harvesting complexes in plants [63]. Sergey Bombin et al. investigated the photosynthetic and related biochemical adaptations of IONPs in soil-grown *Arabidopsis* using a gas exchange system, carbon isotope ratio and chlorophyll content analysis. They observed that enhanced stomatal conductance of *Arabidopsis* promoted photosynthesis and increased biomass by 38% after treatment with a 500 mg/kg concentration. In addition, the uptake of iron by the roots and leaves was increased (Figure 9). Although this may be due to IONPs providing bioavailable iron as a nutrient or increasing phytohormone content and antioxidant enzyme activity, the underlying mechanism is not well-understood [73].

Thus, at the physiological level, the effects of MNPs on plants are more dependent on the particles themselves, for which the plants could have corresponding countermeasures. In contrast, among other effects, ions released by MNPs regulate plant growth and development by altering plant photosynthetic centers and reducing chlorophyll content, but are mostly limited to the high ion concentration range [87].

### 3.3. Toxic Effects at the Molecular Level

Au NPs with different surface charges can become adsorbed to the marginal cells of *Arabidopsis* roots, after which they can affect the growth and development of the plant. Correspondingly, downregulated expression of miR164, miR167, miR395, miR414, miR398 and miR408 in *Arabidopsis* further corroborates their involvement in the plant stress response and the complexity of their regulatory network via regulation of their target genes [62].

Sun et al. found that plants’ response to Ag NPs was mainly associated with transcription, protein degradation, the cell wall, direct DNA/RNA/protein damage and cell division [60]. Ag NPs have not only been shown to activate genes associated with both metal and oxidative stress responses and induce the expression of genes related to the phytohormone abscisic acid (ABA) signaling pathway, but to also inhibit the upregulation of lateral root development growth hormone response genes, and downregulate genes associated with pathogen and hormone signaling responses [50,88]. When *Arabidopsis* roots were exposed to Ag NPs, expressions of homologous recombination (HR)-related genes and the alleviation of transcriptional gene silencing (TGS) in aerial leafy tissues were examined as genotoxic endpoints. It can be seen HR gene expression in aerial leaf tissue was upregulated, and TGS-silenced repetitive elements in aerial tissues could be observed. These observations suggest that the plant systemic response may involve distant induction of Ag NPs’ genotoxicity (Figure 10) [24].

Silver occurs naturally in several oxidation states, of which elemental silver (Ag0) and monovalent silver (Ag^+^) are the two most common states. Previously, it was hypothesized that Ag^+^ release was responsible for Ag0 toxicity [89,90]. However, in a study by Jane et al., who compared the toxic effects of Ag^+^ and Ag NPs on *Arabidopsis*, the authors obtained inconclusive results. Studies on plant metabolism have confirmed that while both Ag NPs and Ag^+^ could induce glycolysis and affect the TCA cycle and aspartate family pathways, there are some metabolic changes (shikimate–phenylpropanoid biosynthesis, tryptophan and galactose metabolism) that occur only upon Ag NPs treatment. By comparing the differences in Ag NPs and Ag^+^ stresses, 111 genes responsible for the response to fungal infection, anion transport and biological functions associated with the cell wall/plasma membrane were found to be unique to Ag NPs at the Arabidopsis genome level (Figure 11) [24].

Due to their different solubility, MNPs have been shown to affect plant growth and development and gene expression by releasing metal ions or attaching around plant tissues. Jin et al. found that high concentrations of Al_2_O_3_ NPs stimulated the transcription of *Arabidopsis* root-development-associated genes and nutrient-related genes to promote root growth, but observed conflicting results when the same concentrations of Al ions were used because Al ions were highly toxic to plant growth and photosynthesis, and caused severe oxidative stress [91].

ZnO NPs can downregulate the expression of microtubulin-related genes in *Arabidopsis* under hydroponic conditions to promote the degradation of microtubulin monomers, which in turn affects the plant’s cell division [92,93,94]. Wu et al. demonstrated that low concentrations of ZnO NPs could increase *Arabidopsis*’s genomic instability by cooperating with other environmental stresses [95]. Under ZnO NPs, stress ethylene downregulated the expression of cell-cycle-related genes and inhibited the growth of *Arabidopsis*. Changes in sugar content, chlorophyll content, DAB and NBT staining and the antioxidant defense system showed that ZnO NPs were toxic to all genotypes of *Arabidopsis* [65,96]. Whether this generalized molecular level response originates from ZnO NPs or the release of Zn^2+^ is a hot topic of current research. Wan et al. compared the transcriptomic response of *Arabidopsis* roots to ZnO NPs, bulk ZnO and ionic Zn^2+^. They observed that the similarity of the transcriptional profiles and the increased number of transcripts with increasing concentration of Zn^2+^ in the culture medium suggested that the release of Zn^2+^ could be contributing to the toxic effects of ZnO NPs on the plant at the molecular level (Figure 12) [97].

Consistent with the effects observed at the physiological level of *Arabidopsis*, CuO NPs by themselves had a much higher molecular level effect than Cu^2+^. After 2 h of exposure to CuO NPs treatment, global gene expression analysis showed much more robust upregulation of oxidative-stress-related genes than with corresponding Cu^2+^ exposure [67].

Treatment of Arabidopsis mutants with TiO_2_ NPs and CeO_2_ NPs altered the regulation of 204 and 142 genes, respectively, and affected a range of metabolic processes, such as DNA metabolism, hormone metabolism, tetrapyrrole synthesis and photosynthesis, in the plant. Although the two nanoparticles differ significantly in the molecular mechanisms they use to promote sprouting growth, they altered oxidative stress reactions, as well as expression of genes that encode for responses to stimuli, localization and growth and development (Figure 13), suggesting the effect of each MNP on altered gene expression could be qualitatively and quantitatively different [98].

ZnSe QDs were also shown to cause oxidative stress in *Arabidopsis* leaves to significantly inhibit the growth of Agrobacterium rhizogenes in the vicinity of *Arabidopsis* roots. Most genes were to be found repressed in roots treated with 100 μM of ZnSe QDs. This was the first time that differential regulatory responses to ZnSe QDs exposure in *Arabidopsis* were observed by gene expression and metabolomic characterization [48]. Compared to Cd^2+^ ions, the regulation of genes related to ABTS and DPPH radicals, total phenols, GSH redox status and lipid peroxidation in plants treated with CdS QDs was not significant, suggesting that this could be the reason why QDs only release Cd^2+^ to a limited extent [99,100].

MNP exposure can affect plant growth via alterations in their gene expressions and metabolic genetics. Studies suggest that the effects of MNPs on *Arabidopsis* at the molecular level are largely consistent with those observed at the normal and physiological levels of uptake and transport in *Arabidopsis*. The effects discovered at the molecular level in *Arabidopsis* are related to the detoxification and transport of different metal elements in vivo, and the regulation of DNA replication, transcription and translation, manifested as the material specificity of MNPs. They can be summarized as the effects of the particle itself, which are considered to be the primary mechanism, and the effects of the release of ions in a way that causes differential kinetics and cytotoxicity, particularly through upregulation or downregulation of the expression levels of certain genes (Table 2) [101].

## 4. Conclusions and Outlook

In summary, MNPs enter *Arabidopsis* via the plastid extracellular pathway but have a low in vivo internalization and transfer rate. Only a few MNPs can be readily transferred to the aboveground parts of *Arabidopsis*. In response to stresses from MNPs, changes in *Arabidopsis* can be observed at the following three levels: the morphological level, represented by inhibition of root and leaf growth and decreased biomass; the physiological level, manifested by reduced chlorophyll content, affected photosynthetic efficiency and others; and the molecular level, comprising upregulation of antioxidant-related genes, upregulation of resistance signaling pathway genes, downregulation of mRNA expression and more. Further, the interactions between MNPs and *Arabidopsis* depend on the surface charge, exposure concentration, particle size and morphology of the MNPs. Most importantly, some MNPs can dissolve in biological fluids, release metal ions that interact with biomolecules and act in conjunction with the material itself to cause redox imbalance in *Arabidopsis*, the two main mechanisms by which MNPs can be toxic to *Arabidopsis*. To more comprehensively explore the effects on *Arabidopsis* and clarify the underlying mechanisms of action, the following should be systematically considered in future studies:(i)Currently available literature that investigated the effects of MNPs on *Arabidopsis* used experimental settings that differed considerably from actual environment settings. For instance, the medium used was water or sandy soil rather than actual soil. The treatment time was comparatively shorter than that observed in nature, and *Arabidopsis* was exposed in a relatively single manner to MNPs, i.e., the roots were mainly exposed to the soil in the presence of MNPs, and thus, the effects of using leaf sprays or hydroponics remain to be determined.(ii)The physicochemical properties, treatments and growth stages of the MNPs used in the experiments differed among studies. Therefore, the source of nanomaterials, preparation methods, testing equipment selection and design of exposure conditions should be standardized.(iii)The internalization of metal ions produces different levels of toxicity to plants than direct ingestion of metal ions. The toxic effect seems to be concentration-dependent. Further, as the toxic effects from MNPs could be from the released metal ions alone or the combined effect of nanoparticles themselves, the source of toxicity should be identified.(iv)As studies at different levels of the plant’s response, i.e., subcellular, physiological and biochemical levels, are being performed, the interactions between MNPs and *Arabidopsis* should be designed in a way that combines the traditional toxicological research methods with histological techniques (transcriptomics, metabolomics and proteomics) to provide more accurate and in-depth elucidation of the mechanisms of MNP-based phytotoxicity.

## Figures and Tables

**Figure 1 materials-15-04539-f001:**
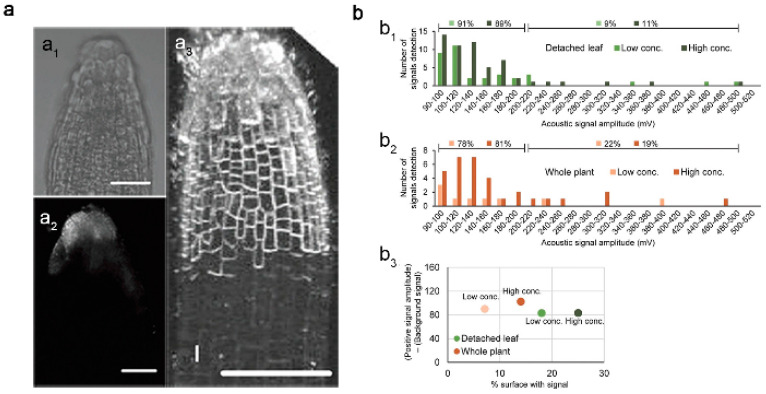
(**a**) Localization of 40 nm silver nanoparticles (Ag NPs) in Arabidopsis roots. (**a_1_**) Two-week-old control root tip demonstrating no Ag NP signal. (**a_2_**) 267.36 mg/L of Ag NPs, 1 week. Ag NPs are shown in the columella cells as an illuminating white crown. (**a_3_**) A surface overview of a brown root tip. (**b**) Statistical analysis of acoustic signals detected from GNPs in Arabidopsis leaves. (**b_1_**,**b_2_**) The frequency of leaf signal amplitudes is compared between (**b_1_**) high- and low-GNP-concentration exposure to detached leaf petioles and (**b_2_**) high- and low-GNP-concentration exposure to whole plants for two different durations. Signal amplitudes below 200 mV and above 200 mV are indicated on upper side of each graph. (**b_3_**) Percentage of leaf surface that emitted detectable signal (% surface with signal, x axis) and acoustic signal amplitude (average signal amplitude over 90 mV—average signal amplitude below 90 mV, y axis) from (**b_1_**) and (**b_2_**) are plotted. Detached leaf data are shown in green; whole-plant exposure data are shown in orange. Reprinted with permission from Refs. [27,28].

**Figure 2 materials-15-04539-f002:**
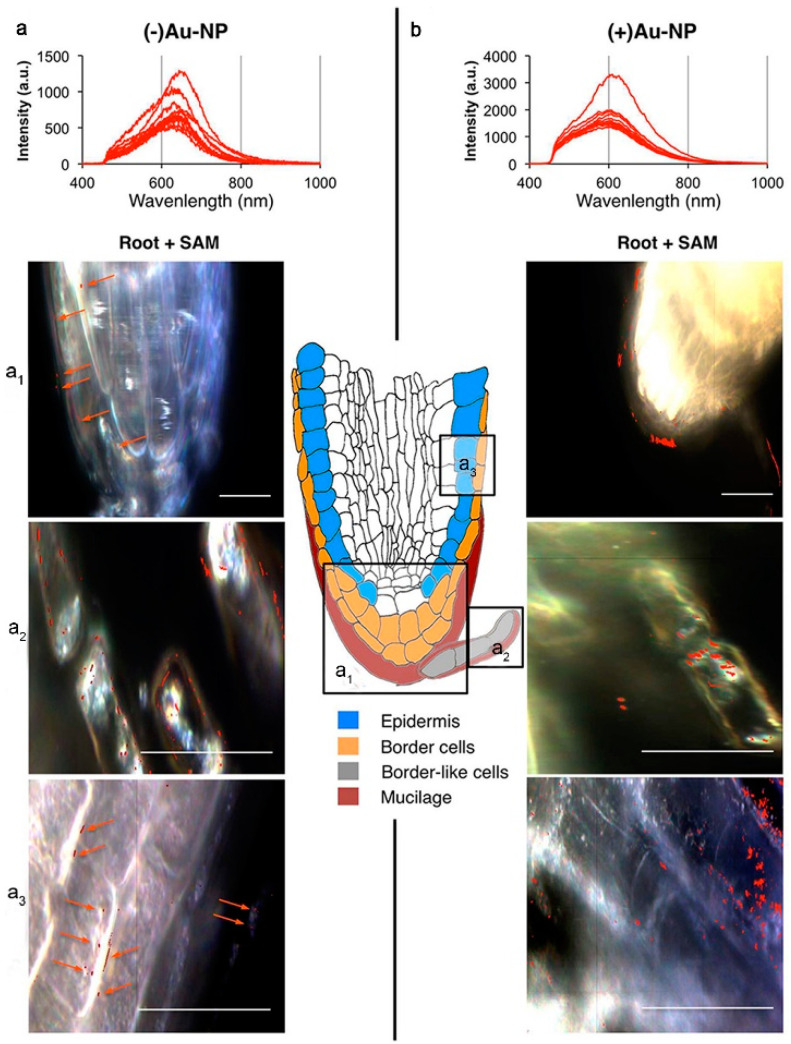
Spectral libraries used for the nanomaterial mapping of (**a**) (−) Au-NPs and (**b**) (+) Au-NPs. (**a_1_**–**a_3_**) Dark-field microscopy images of *Arabidopsis thaliana* roots exposed to 10 mg/L of (−) Au-NPs (**left**) and (+) Au-NPs (**right**). Red pixels: (−/+) Au-NPs mapped using the spectral angular mapping algorithm (SAM; 0.085 rad). Images of different root compartments in the top root. (**a_1_**) Root cap with border-like cells and mucilage. (**a_2_**) Detaching border-like cells. (**a_3_**) Lateral root cap and epidermis. (The orange arrows points to where the Au-NPs are distributed.) Reprinted with permission from Ref. [29].

**Figure 3 materials-15-04539-f003:**
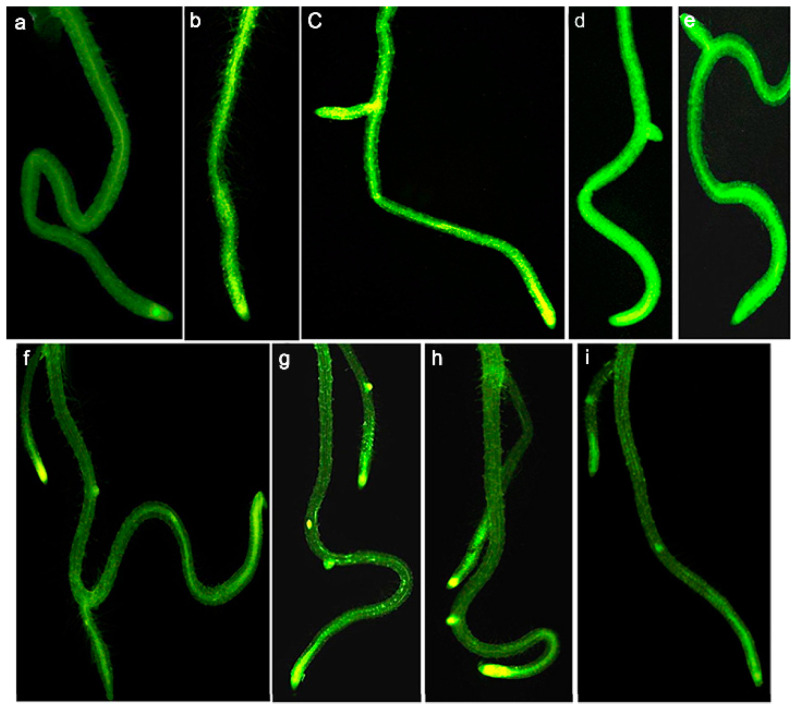
Accumulation of zinc in roots of *A. thaliana* seedlings evidenced by Zynpyr-1fluorescence after exposure to various concentrations of zinc and ZnO NPs. (**a**) Control and seedlings grown in the presence of 20, 50, 100 and 200 mg/L of (**b**–**e**) Zn and (**f**–**i**) ZnO NPs. Reprinted with permission from Ref. [36].

**Figure 4 materials-15-04539-f004:**
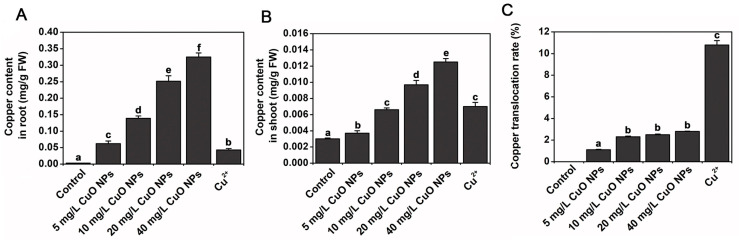
Effect of CuO NPs and Cu^2+^ on copper uptake and transfer. (**A**,**B**) Effect of CuO NPs (0–40 mg/L) and Cu^2+^ (1.4 mg/L) on copper accumulation in roots and shoots. (**C**) Effect of CuO NPs (0–40 mg/L) and Cu^2+^ (1.4 mg/L) on copper transfer in roots and shoots. Lowercase ‘a to f’ indicated the significant different *p* < 0.05 in histogram. Reprinted with permission from Ref. [37].

**Figure 5 materials-15-04539-f005:**
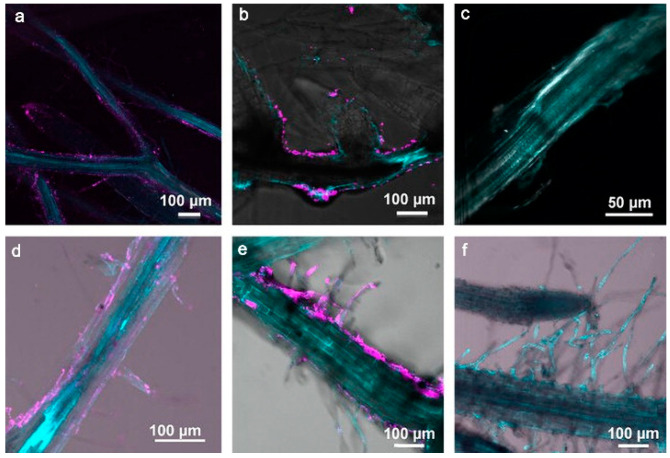
Superposition of fluorescence and light microscopy images of plants’ roots from exposure to QD suspensions in Hoagland’s solution (HS) for (**a**) 1 day and (**b**) 7 days, and HS + humic acids (HAs) for (**d**) 1 day and (**e**) 7 days. Images of unexposed plants in (**c**) HS and (**f**) HS + HA are also provided for comparison. QD emission is shown in pink. Endogenous emission is shown in blue-green. Reprinted with permission from Ref. [49].

**Figure 6 materials-15-04539-f006:**
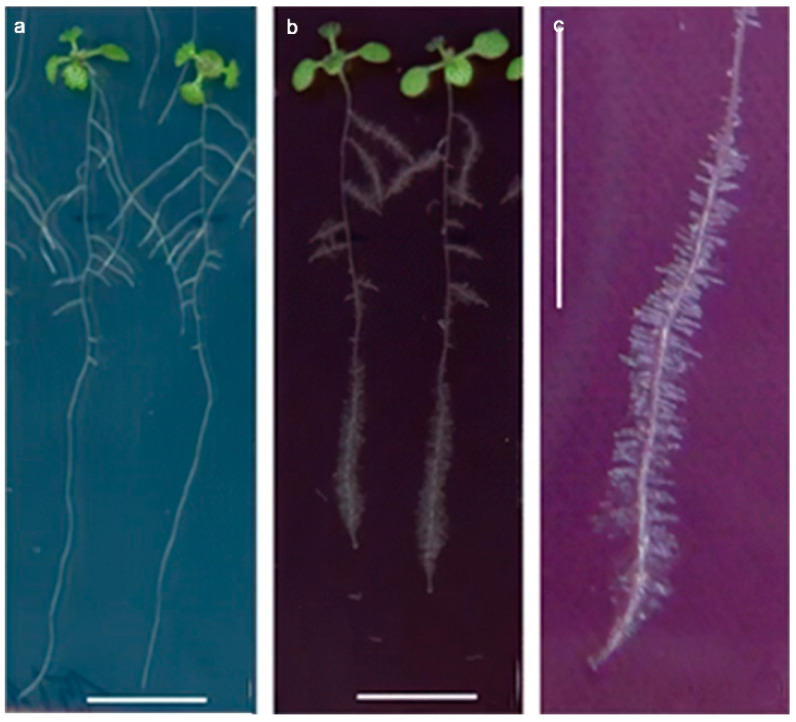
Effect of different concentrations of 10 nm Au NPs on root hair growth in *A. thaliana* seedlings. (**a**) Control. (**b**) Treated with 100 mg/L of Au NPs. (**c**) Induced root hair growth in plants exposed to 100 mg/L of Au NPs. Reprinted with permission from Ref. [61].

**Figure 7 materials-15-04539-f007:**
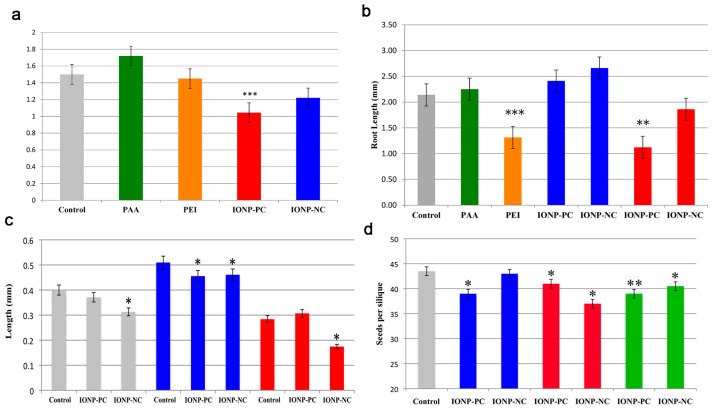
(**a**) Effect of IONP treatment on overall seedling length in *A. thaliana*. Treatments marked with asterisks were significantly different from the control with *p* < 0.0001 (**b**) Changes in *A. thaliana* seedling root length after exposure to IONPs. Treatments with significant difference of *p* < 0.009 are marked with two asterisks; *p* < 0.0001 are marked with three asterisks. (**c**) Effects of IONPs on pollen tube growth in *A. thaliana.* Treatments with significant differences from the control are marked with an asterisk (*p* < 0.05). (**d**) Treatment of *A. thaliana* with IONPs resulted in reduced seed production. Single asterisks represent treatments that were significantly different from the control with a *p* < 0.05; two asterisks indicate treatments that were significantly different with a *p* < 0.01. Reprinted with permission from Ref. [73].

**Figure 8 materials-15-04539-f008:**
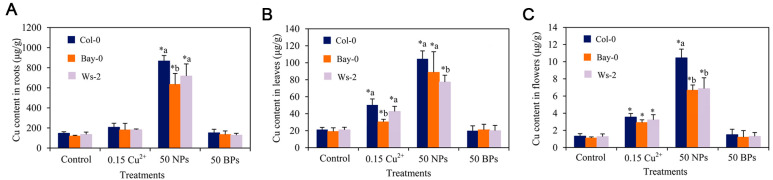
Cu contents in (**A**) roots, (**B**) leaves and (**C**) flowers of different *Arabidopsis* ecotypes after treatment with distilled water (control), 0.15 mg/L Cu^2+^ ions (0.15 Cu^2+^), 50 mg/L CuO NPs (50 NPs) and 50 mg/L CuOBPs (50 BPs). Significant difference among different treatments compared control was marked with “*”. For a given treatment, different letters represent significant differences among different ecotypes (*p* < 0.05, LSD, n = 3). Reprinted with permission from Ref. [38].

**Figure 9 materials-15-04539-f009:**
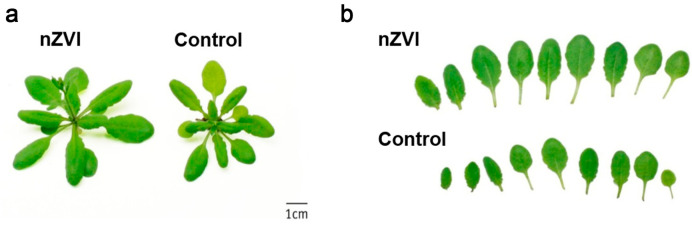
Phenotype (**a**) and photo (**b**) of growth of control and nanoscale zerovalent iron (nZVI)-exposed *Arabidopsis* shoot at 21 days. Reprinted with permission from Ref. [86].

**Figure 10 materials-15-04539-f010:**
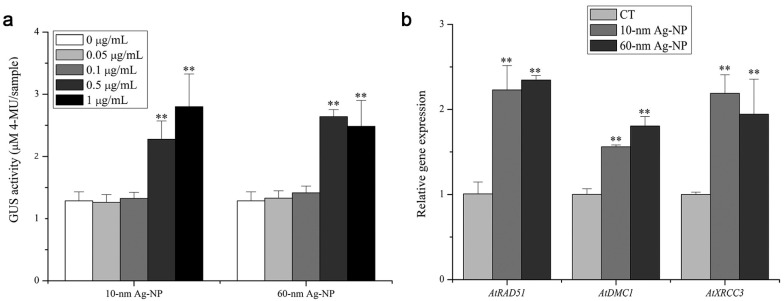
The effect of root exposure to Ag-NP on the expression of HR-related genes. (**a**) The A. thaliana line L5-1, which harbors a single insert of a multicopy of P35S: GUS (TGS-GUS), is presented by Dr. Ortrun Mittelsten Scheid. GUS activity in the aerial tissues of 15-6# plants, 7 days after root exposure to Ag-NP. (**b**) The mRNA level of other HR-related genes in the aerial tissues of wild-type plants, 7 days after root exposure to 1 μg/mL of Ag-NP. Results are the means ± SD (n ≥ 12 for GUS activity; n = 3 for RNA level, *t*-test ** *p* < 0.01). Reprinted with permission from Ref. [55].

**Figure 11 materials-15-04539-f011:**
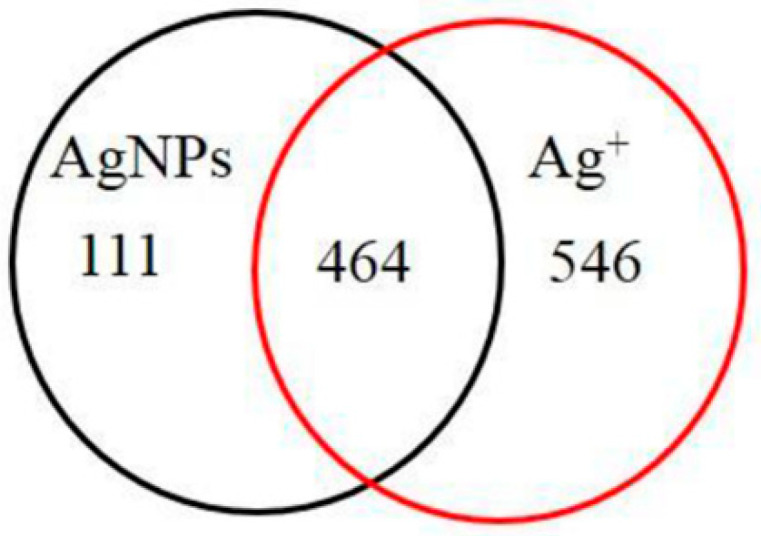
Venn diagrams of genes with more than two-fold expression changes and shared among the six stresses between Ag NPs and Ag^+^. Reprinted with permission from Ref. [24].

**Figure 12 materials-15-04539-f012:**
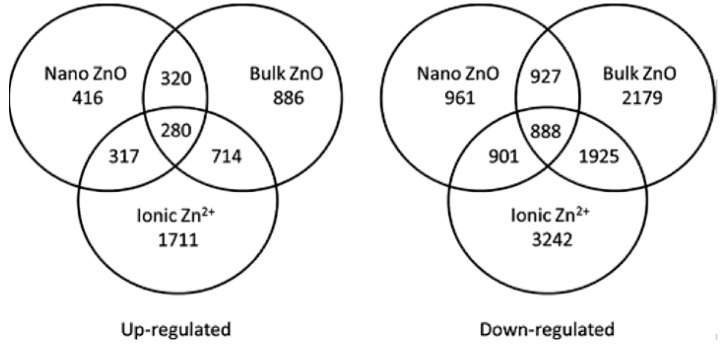
Numbers of up- and downregulated genes after ZnO NPs, bulk ZnO and ionic Zn^2+^ exposure. Reprinted with permission from Ref. [66].

**Figure 13 materials-15-04539-f013:**
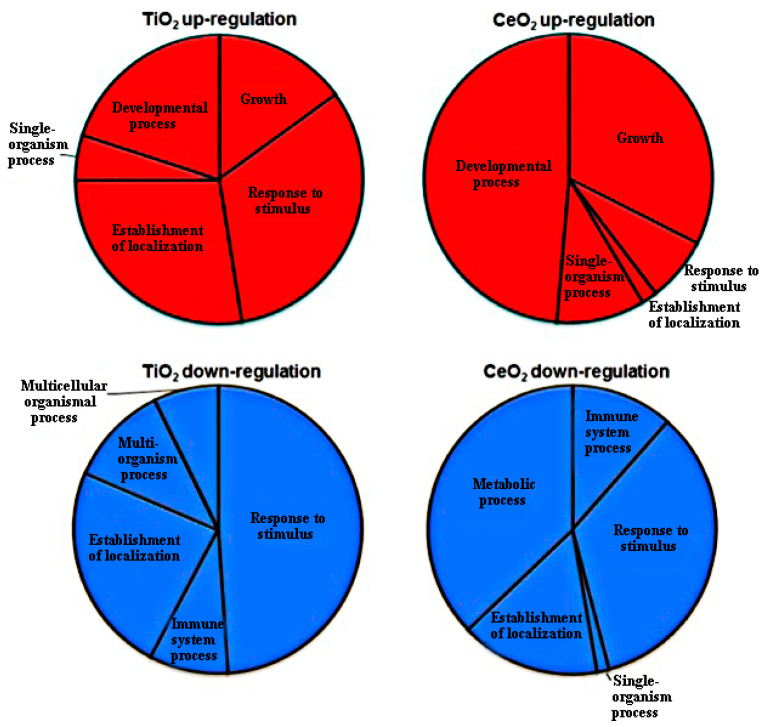
Pie charts of top 5 categories of upregulated and downregulated genes in *Arabidopsis thaliana* germinants exposed to nano-titania and nano-ceria annotated to broad functions. Reprinted with permission from Ref. [82].

**Table 1 materials-15-04539-t001:** Toxic effects of various MNPs in *Arabidopsis*.

MNPs	Size	Concentration	Impact	Reference
AgNPs	20, 40, 80 nm	7.0 × 10^10^,	Inhibited seedling root elongation and showed a linear dose–response relationship.	[28]
		9 × 10^9^,		
		1.1 × 10^9^		
		particles/mL		
AgNPs	41 ± 1.5 nm	Greater than 300 mg/L	The inhibitory effect was saturated at 3000 mg/L, inhibiting growth and photosynthetic efficiency.	[54]
AgNPs	10, 60 nm	0, 0.05, 0.1, 0.5 and 1 μg/mL	After exposure to 60 nm Ag NPs, the Ag content in the aerial tissues was significantly increased.	[55]
AgNPs	10 nm	0.02 mg/L	Ag NPs no longer existed as intact individual particles but were aggregated and/or biotransformed in the plant.	[56]
AgNPs	10 nm	1.0, 2.5 mg/L	Induced glycolysis and affected the TCA cycle and aspartate family pathway. Glycine, serine and threonine metabolism were reduced.	[57]
AgNPs	25.6 ± 5.1 nm	0, 10, 30, 40, 50, 60, 70, 80, 90, 100, 150, 200 and 300 mg/L	Low Ag NPs levels induced ROS, accelerated root tip cell proliferation and promoted root growth. Relatively high concentrations of Ag NPs inhibited cell division, thereby limiting root growth.	[58]
AgNPs	10–12 nm	12.5 mg/kg	Affected the quality of pod and the growth of offspring seed, delayed flowering time by altering relevant pathways (photoperiod, autonomous and vernalization pathways) and inhibited pollen formation and development. Any negative effects on flower development could be transferred to the offspring.	[59]
PVP-coated AgNPs	25 nm	10, 30, 50, 100, 150 mg/L	Suppression of root to gravity with dose-dependent effects.	[60]
Au NPs	60 nm	1 ppm	Upon entering the leaves, it acted as a photothermal agent and remotely activated local biological processes in the plant on demand.	[27]
Au NPs	5 nm	25 µg/mL	Different surface charges affected *Arabidopsis* root development.	[30]
Au NPs	13.4 ± 1.3, 12.1 ± 0.8 nm	10 mg/L	Separated border-like cell sheets (isolated from the root) and associated mucus accumulated and trapped NPs independent of particle charge, in contrast to the marginal cells on the root crown that exhibited charge specificity.	[29]
Au NPs	10–18 nm	100 mg/mL	Au NPs had significant effects on the lateral roots of *Arabidopsis*. At the highest concentration, the minimal Au NPs inhibited the length of primary roots but contrarily also promoted the growth of hairy roots.	[61]
Au NPs	24 nm	10, 80 µg/mL	Exposure to Au NPs at 24 nm at concentrations of 10 and 80 μg/mL significantly increased seed germination, nutritional growth and free-radical scavenging activity.	[62]
CdSe/ZnS QDs	6.3 ± 0.7 nm	5 µg/mL	The ratio of reduced glutathione (GSH) to oxidized glutathione (GSSG) was reduced in the plants.	[49]
ZnSe QDs	-	100, 250 μM	Caused oxidative stress in the leaves.	[48]
CuZn NPs	20–30 nm	30 mg/L	The photosystem II (PSII) function of young leaves was negatively affected.	[63]
ZnO NPs	30 nm	0.16–100 mg/L	High doses of ZnO NPs resulted in upregulation of the stress hormone abscisic acid, mainly in the apical regions and leaves.	[64]
ZnO NPs	30 nm	50, 100, 200 and 300 mg/L	Caused sugar and chlorophyll changes, DAB and NBT staining and antioxidant defense systems.	[65]
ZnO NPs	20–45 nm	0, 20, 50, 100, 200 mg/L	*Arabidopsis*’s fresh weight and primary root length were reduced, except when at a concentration of 20 mg/L.	[36]
ZnO NPs	20 nm	4 mg/L	There were 816 upregulated transcripts and 2179 downregulated transcripts.	[66]
CuO NPs	40 nm	10, 20, 40 mg/L	Interference with dynamic changes in actin led to abnormal apical cell development and inhibition of growth hormone transport, causing secondary damage to plant cells.	[37]
CuO NPs	30–50 nm	10, 20 mg/L	After 10 and 20 mg/L treatment for 2 h, the root cells of *Arabidopsis* were severely damaged.	[67]
CuO NPs	38 nm	50, 100, 200, 300, 400 mg/L	Affected rosette size, biomass, chlorophyll content, lipid peroxidation, ROS accumulation and cellular ultrastructure in *Arabidopsis*.	[68]
CuO NPs	20–40 nm	20, 50 mg/mL	The growth of *Arabidopsis* seedlings of different ecotypes (Col-0, Bay-0 and Ws-2) and the germination of their pollen and harvested seeds were inhibited.	[38]
CuO NPs	-	5, 10 μg/mL	Elevated endogenous H_2_S and Cys content inhibited *Arabidopsis* root elongation in a dose-dependent manner.	[69]
CuO NPs	-	10, 20 μg/mL	Strongly inhibited the growth of *Arabidopsis*.	[67]
TiO_2_	5–15 nm	0.1, 0.5 mM	Showed phytotoxicity and could induce autophagy and protect plant cells from nanoparticle-induced damage, especially oxidative damage to chloroplasts.	[70]
TiO_2_	5–15 nm	50, 100 mg/L	Reduced TC toxicity and increased the expression of both γ-glutamyl cysteine synthase (ECS) and glutathione synthase (GS) in *Arabidopsis*.	[71]
CeO_2_	10–30 nm	0–2000 ppm	Exposure to CeO_2_ NPs at 250 ppm significantly increased plant biomass. At 500–2000 ppm CeO_2_ NPs, plant growth was reduced by up to 85% in a dose-dependent manner, and chlorophyll production was reduced by nearly 60% and 85% at 1000 and 2000 ppm, respectively. At 1000 ppm, MDA formation was increased by 2.5-fold.	[42]
CeO_2_	15–30 nm	100, 200, 500, 1000, 2000 and 3000 mg/L	High concentrations of CeO_2_-NPs inhibited plant growth and adversely affected plants’ antioxidant system and photosystem.	[72]
In_2_O_3_	20–70 nm	0–2000 ppm	Resulted in a 3.8–4.6-fold increase in glutathione synthetase (GS) transcription products.	[42]
Fe_2_O_3_	30 nm	3, 25 mg/L	The 3 mg/L treatment had no significant effect on seedling and root length, and the 25 mg/L treatment resulted in a reduction in seedling and root length.	[73]

**Table 2 materials-15-04539-t002:** Gene expression of *Arabidopsis thaliana* exposed to different MNP treatments.

Types of MMNPs	Exposure Concentration	Exposure Time	Experimental Results	Reference
Ag	5 mg/L	10 d	Raise: respond to abiotic stress (mental stress, oxidation stress, salt stress, osmotic stress, hunger stress and water stress)	[55]
			Lower: respond to pathogen stress and hormone stimulation (abscisic acid, auxin and ethylene)	
Ag	12.5 mg/L	45 d	Raise: organic acids, sugars, amino acids	[24]
			Lower: amino acids, phenols	
Au	100 g/L	-	Lower: expression of miR164, miR167, miR395, miR414, miR398 and miR408	[62]
ZnO	4 mg/L	7 d	Raise: respond to abiotic stress (oxidative stress, salt stress, osmotic stress and water stress) and biological stress (pathogen defense), and participate in Zn^2+^ binding, transport and steady state	[65]
			Lower: participate in cell tissue and biogenesis (tubulin, arabinogalactan glycoprotein), DNA or RNA metabolism (histone)	
ZnO	100 mg/L	7 d	Raise: lateral roots develop in response to abiotic stress (oxidative stress, salt stress, osmotic stress and water stress) and biological stress (wound stimulation and pathogen defense)	[36]
			Lower: participate in cell tissue and biogenesis (translation, nucleosome assembly, tubulin), electron transfer	
CuO	10 mg/L	7 d	Raise: response to abiotic stress and biotic stress, Cu^2+^ binding and transport, plant hormone signal transduction	[38]
			Lower: participation in metal homeostasis and transport and root hair development	
TiO_2_	50 μg/mL	7 d	Raise:	[70]
			Small particle size: increased expression level of AtRAD54	
			Big particle size: unable to increase AtRAD54 expression	
TiO_2_	20 μg/mL	7 d	Raise:	[82]
			Small particle size: AtRAD51, AtDMC1, AtXRCC3	
			Big particle size: AtRAD51, AtDMC1, AtXRCC3	
CeO_2_	1 g/L	-	Raise: glutathione metabolism (oxidative stress response) and metal stress response genes	[102]
Al_2_O_3_	10 mg/L	10d	Raise: respond to abiotic stress and biotic stress, cell wall development, nitrogen and phosphorus transport and root development	[91]

## Data Availability

The data used in this research have been properly cited and reported in the main text.

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
