# Peer review of "Recent Advances in Metal-Based Nanoparticle-Mediated Biological Effects in *Arabidopsis thaliana*: A Mini Review"

_materials, 2022, doi:10.3390/ma15134539_

Round 1

Reviewer 1 Report

The authors systematically reviewed the effects of MNPs on Arabidopsis thaliana. This mini review helps researchers who attend to realize a use of MNPs for engineerings, medicals, or environments, evaluating harmful impacts on global environments. However, the figures which authors introduced were not clearly explained, and thus, readers will not understand and not be convinced.

P.2

The surface modification of Ag NPs and GNPs should be clearly stated.

P.2 line 83 “In contrast … ”

This sentence does not explain Fig. 1b correctly.

Fig. 2

What the orange allows indicate?

P.4 line 123 “They detected…”

In Fig. 3, I see strong fluorescence when treated with Zn but ZnO NPs, which is opposite to the authors’ description.

Fig. 5

What are HS and HA?

Table 1

The diameter description for TiO2 NPs (ref. [66]) is wrong.

Fig. 8

What does the legend of graphs represent? What is BP?

Fig. 9

What is nZVI?

Fig. 10

What is GUS? What do the genes represent?

Fig. 13

What do the C and T mean?

I cannot see that “they alter a similar suite of genes via different pathways and processes associated with growth promotion”

Author Response

Response to Reviewer #1:

Q1: P.2 The surface modification of Ag NPs and GNPs should be clearly stated.

Response:

Thanks for your valuable opinion. We have indicated the modified nanoparticles in the article,you can see it separately on page 3 at lines 119-121.

Q2: P.2 line 83 “In contrast …” This sentence does not explain Fig. 1b correctly.

Response:

Thank you for your suggestion, by further interpreting this manuscript here, we have replaced the Fig. 1 b and explained it in the text(On page 2, at line 88-91. On page 3, at line 92-93).

Q3: Fig. 2 What the orange arrows indicate?

Response:

Thank you for your suggestion, the orange arrows specifically represent the content we have marked in the text you can see it on page 4 at lines133.

Q4: P.4 line 123 “They detected…”

In Fig. 3, I see strong fluorescence when treated with Zn but ZnO NPs, which is opposite to the authors’ description.

Response:

Thank you for your valuable comments. We have reinterpreted it in the text based on your suggestions, you can see it on page 4 at lines146-148.

Q5: Fig. 5 What are HS and HA? (196-198)

Response:

Thank you for your valuable comments, we added Hoagland’s solution (HS) and HS + Humic acids (HA) on page 6, at line 217-218.

Q6: Table 1 The diameter description for TiO2 NPs (ref. [66]) is wrong.

Response:

We are sorry for our negligence, we have modified it in the table, we change 100mg/mL to 10-15nm.

Q7: Fig. 8 What does the legend of graphs represent? What is BP?

Response:

Because of our negligence, it is not fully expressed,we have added the legend of graphs represent to the original manuscript, and explained what is BP on page12, at line 301.

Q8: Fig. 9 What is nZVI?

Response:

Because of our negligence, it is not fully expressed,We have explained what is nZVI on page 13, at line 343.

Q9: Fig. 10 What is GUS? What do the genes represent?

Response:

Thank you for your valuable comments, we have explained what is GUS and what do the genes represent on page 13, at line 372-375.We also marked what the genes represent in lines 364-366.

Q10: Fig. 13

What do the C and T mean? ( )

I cannot see that “they alter a similar suite of genes via different pathways and processes associated with growth promotion”

Response:

I'm sorry that due to our wrong understanding, the picture can not accurately reflect the content of the text, we replaced Fig. 13 to explain the case “they alter a similar suite of genes via different pathways and processes associated with growth promotion” in a different way.

Reviewer 2 Report

It is a very good study with overall adequate presentation. Some additions are needed:

1) Authors should further emphasize on the novelty of their work.

2) Some minor typos, grammar and syntax errors should be carefully revised and corrected accordingly.

3) Reference can be even more updated (more recent relative works).

Author Response

Response to Reviewer #2:

Q1: Authors should further emphasize on the novelty of their work.

Response:

Thank you for your thoughtful suggestion. We add to this content in lines 57-58 on page 2 of the text to emphasize on the novelty of our work.

Q2: Some minor typos, grammar and syntax errors should be carefully revised and corrected accordingly.

Response:

Thank you for your careful review of our article, for which we are deeply grateful. We have gone over it again and corrected the minor typos, grammar and syntax errors, such as line 10,18,34,42 and 45 on page 1 and so on.

Q3: Reference can be even more updated (more recent relative works).

Response:

Thank you for your thoughtful suggestion. We have added references about more recent relative works, you can see them on page 5 at lines 175-178, page 6 at lines 203-204, page 7 at lines 225-226 and 230-236, page 10 at lines 244, page 14 at lines 407, page 17 at lines 447.

Reviewer 3 Report

The paper presents a very interesting and documented study on the effect of metal-based nanoparticles on growing plants in the soil especially with respect to the aspects of nano-biosafety. The potential impact of nanoparticles on the growing soil environment in a model plant has been thoroughly investigated.  The model plant chosen to study the impact of environmental stress on plants growth is well described since this model plant adapt to the stresses that are similar in many other important plants. Therefore conclusions obtained from the scientific research can be used as a reference for other plants behavior in what concerns  the mechanisms through which nanomaterials affect plant growth and development.

The paper is of good quality and is recommended for publication in the journal Materials.

Author Response

Response to Reviewer #3:

Thank you very much for your comments and endorsement of the manuscript. In the revised manuscript, we have humbly accepted the suggestions of the reviewers and carefully revised any shortcomings and errors. We would like to thank all the reviewers for their understanding and recognition during this process. We do feel that the current revised version is a substantial improvement over the original manuscript.

Reviewer 4 Report

The review articles deals with the effect of metal-based nanoparticles on the environment, specifically on the growth and development of Arabidopsis thaliana plant as al model plant.  The review includes previous studies on adsorption of the widely studied metal nanoparticles, metal oxide nanoparticles, and CdSe/ZnSe quantum dots by Arabidopsis thaliana plant tissue and their phytotoxic effects. The review includes also previous studies of the effect of toxicity of the aforementioned nanoparticles at the morphological, physiological, and molecular level of the Arabidopsis thaliana plant.

The review is well-organized and covered the subject in an efficient and interesting way to the readers. The conclusions and outlook parts were very well harmonized with the previous finding mentioned in the review.

I recommend publication in the current form.

Author Response

Response to Reviewer #4:

Thank you very much for your comments and endorsement of the manuscript. In the revised manuscript, we have humbly accepted the suggestions of the reviewers and carefully revised any shortcomings and errors. We would like to thank all the reviewers for their understanding and recognition during this process. We do feel that the current revised version is a substantial improvement over the original manuscript.

Reviewer 5 Report

I must say I have no objections to this review. The topic is interesting, and the authors really covered everything in a well-organized and sectioned review paper, even highlighting future research directions. My recommendation is to accept it in its current form.

Author Response

Response to Reviewer #5:

Thank you very much for your comments and endorsement of the manuscript. In the revised manuscript, we have humbly accepted the suggestions of the reviewers and carefully revised any shortcomings and errors. We would like to thank all the reviewers for their understanding and recognition during this process. We do feel that the current revised version is a substantial improvement over the original manuscript.